EMBO
reports

# scientific report

# A transient reversal of miRNA-mediated repression controls macrophage activation

*Anup Mazumder[1], Mainak Bose[1], Abhijit Chakraborty[2], Saikat Chakrabarti[2] & Suvendra N. Bhattacharyya[1+]*

[1]RNA Biology Research Laboratory, Molecular and Human Genetics Division and [2]Structural Biology and Bioinformatics Division, CSIR-Indian Institute of Chemical Biology, Kolkata, India

**In mammalian macrophages, the expression of a number of cytokines is regulated by miRNAs. Upon macrophage activation, proinflammatory cytokine mRNAs are translated, although the expression of miRNAs targeting these mRNAs remains largely unaltered. We show that there is a transient reversal of miRNA-mediated repression during the early phase of the inflammatory response in macrophages, which leads to the protection of cytokine mRNAs from miRNA-mediated repression. This derepression occurs through Ago2 phosphorylation, which results in its impaired binding to miRNAs and to the corresponding target mRNAs. Macrophages expressing a mutant, non-phosphorylatable AGO2—which remains bound to miRNAs during macrophage activation—have a weakened inflammatory response and fail to prevent parasite invasion. These findings highlight the relevance of the transient relief of miRNA repression for macrophage function.**

Keywords: Ago2 phosphorylation; lipopolysaccharide; macrophage activation; miRNA; translational repression

## INTRODUCTION

MicroRNAs (miRNAs), the 21–22 nt long non-coding RNAs, by binding target messenger RNAs (mRNAs) with imperfect complementarities, repress protein synthesis [1]. Different miRNAs control immune response in mammals by coordinating the genesis and maintenance of different humoral cells including macrophages [2]. Macrophages act to clear the invading pathogens primarily by inducing the expression of proinflammatory cytokines and nitric oxide production [3]. miRNAs control cytokine expression in humoral cells to stop uncontrolled cytokine production-related cell death [4] and ensure an optimal inflammatory response [5].

Several reports have shown previously that miRNA levels might be altered by bacterial lipopolysaccharide (LPS) in monocytes and macrophages [6]. LPS binds to Toll-like receptor 4 (TLR4) and induces expression of proinflammatory cytokines including IL-1β, IL-6, tumor necrosis factor alpha (TNF-α) [7]. On the contrary, the protozoan parasite *Leishmania donovani* evades this proinflammatory immune response and induces expression of IL-10, an anti-inflammatory cytokine, to ensure its survival inside the macrophage [8]. LPS stimulation of monocytes and macrophages is accompanied by an increase in miR-155, miR-146 and let-7e and decrease of miR-125b expression in mouse macrophage cells [6,9]. Interestingly, LPS does not cause any change in the expression of majority of the miRNAs including let-7a in murine macrophages [10]. A number of cytokine mRNAs that are upregulated in activated macrophage are targets of these 'unchanged' miRNAs, for example, IL-6 mRNA is a target of let-7a [11]. On activation of macrophage with LPS, translation of the cytokine mRNAs get enhanced. But how the miRNA-targeted cytokine mRNAs become immune to miRNA-mediated repression in LPS-stimulated macrophage is currently unknown.

In the present study, we examined the miRNA activity in LPS-stimulated murine macrophage like cell line RAW 264.7. We found miRNA activity was downregulated at initial period of LPS stimulation. At early hours, LPS treatment induced phosphorylation of protein Ago2 to dissociate miRNA from the active miRNP complex leading to derepression and enhanced protein synthesis from the miRNA-targeted mRNAs including the mRNAs encoding proinflammatory cytokines. This downregulation of miRNA activity at an early period of LPS stimulation was transient and reverted to normal repression level after prolonged LPS exposure. This ensures the optimal proinflammatory response whereas defective derepression of miRNA-targeted proinflammatory cytokine mRNAs at initial hours of macrophage activation leads to poor resistance of host macrophages against invasion of the pathogenic parasite *L. donovani*.

## RESULTS AND DISCUSSION
### Low miRNA activity in LPS treated RAW264.7 cells

mRNA levels of proinflammatory cytokines IL-1β, IL-6, IL-4 and TNF-α were all increased several fold and were at peak at 4 h after LPS treatment whereas the anti-inflammatory cytokine IL-10 level was largely unchanged at 4 h of LPS stimulation (Fig 1A). IL-6 is a let-7a-regulated mRNA and inhibition of let-7a resulted in increased IL-6 mRNA level in RAW 264.7 cells (Fig 1B). IL-6

[1]RNA Biology Research Laboratory, Molecular and Human Genetics Division,
[2]Structural Biology and Bioinformatics Division, CSIR-Indian Institute of Chemical Biology, Kolkata 700032, India
[+]Corresponding author. Tel: +91 33 2499 5783; Fax: +91 33 2473 5197;
E-mail: sb@csiriicb.in or suvendra@iicb.res.in

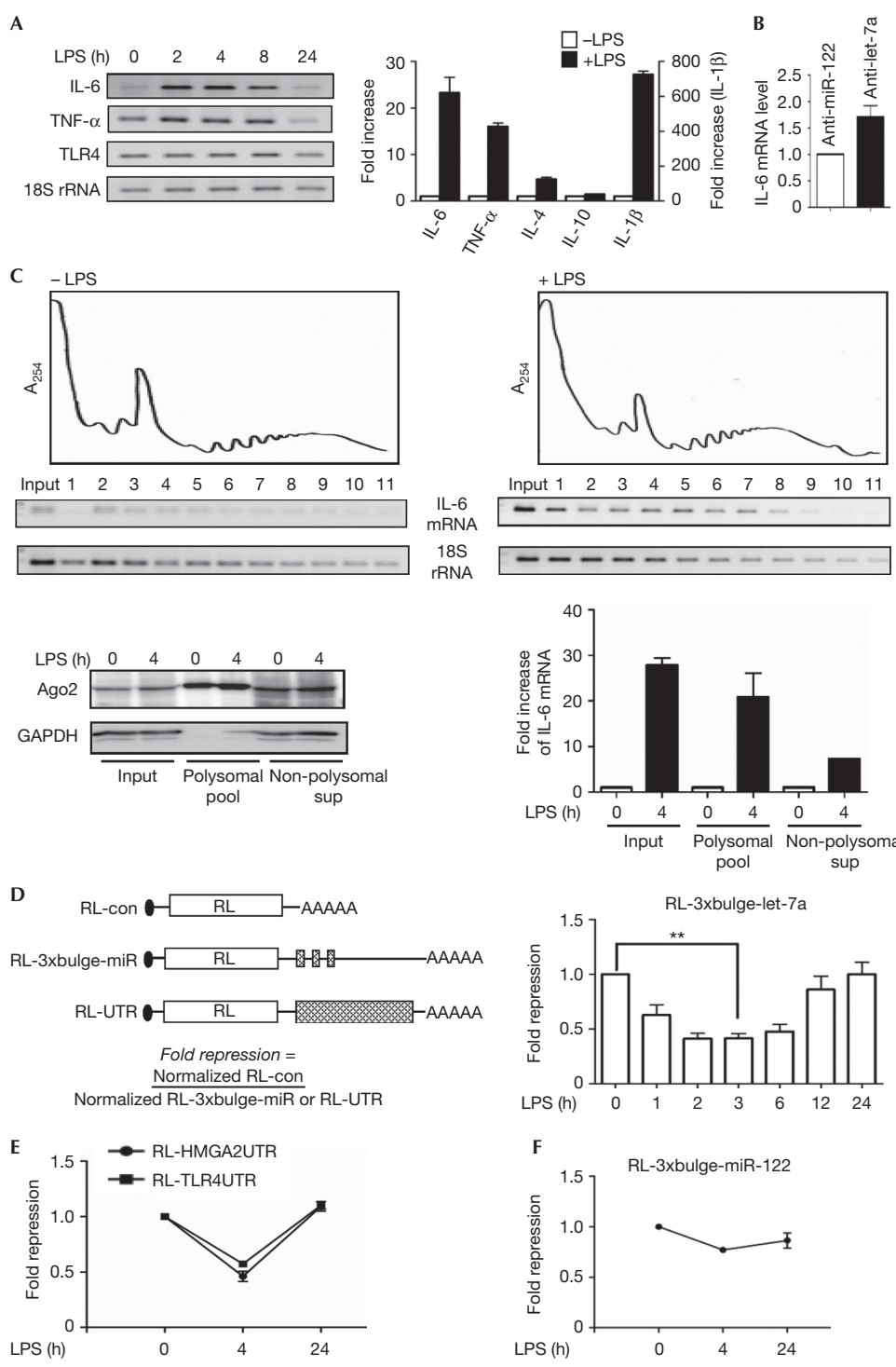

**Fig. 1** | For caption please see page 1010.

showed almost 20-fold increase in RNA level during the early hours of LPS stimulation (Fig 1A). Increased association of IL-6 mRNA with polysomal fractions confirmed an impaired repression of this mRNA by miRNA let-7a in LPS-activated RAW 264.7 cells (Fig 1C). Does this derepression also happen with other miRNA-targeted messages during the early phase of LPS

stimulation? To test, we transfected RAW 264.7 cells with a *Renilla* luciferase (RL) reporter having three imperfect let-7a binding sites (RL-3xbulge-let-7a; described and characterized earlier [12]). Derepression of this let-7a reporter was also noted at early time of LPS treatment but repression was reestablished after prolonged incubation with LPS (Fig 1D and supplementary

# scientific report

◄ **Fig 1 |** LPS treatment induces immediate upregulation of proinflammatory cytokine mRNAs and downregulation of miRNA activity in RAW 264.7 cells. (**A**) Expression of proinflammatory cytokine mRNAs after exposure to LPS. Semi-quantitative (left panel) or real-time RT–PCR (right panel) was done. 18S rRNA served as control. For right panel, RNA from 4-h LPS-treated cells was used. (**B**) Effect of let-7a inhibition on IL-6 mRNA. mRNA was measured by quantitative RT–PCR in RAW 264.7 cells transfected either with anti-let-7a or anti-miR-122 oligos. (**C**) Sucrose density gradient analysis of LPS-stimulated and naïve RAW 264.7 cell extract and semi-quantitative RT–PCR to detect the presence of IL-6 mRNA and 18S rRNA of different fractions (upper panel). Change in IL-6 mRNA level in the polysomal and non-polysomal fractions separated on a 30% sucrose cushion (lower panel). (**D**) Reversal of miRNA-mediated repression in LPS-treated RAW 264.7 cells. In the left panel, schematic representation of different RL reporter constructs used to score miRNA activity. In the right panel, changes in let-7a-mediated repression during LPS treatment in RAW 264.7 cells transfected with luciferase reporters (**$*P<0.0055$). (**E**) Changes in let-7a-mediated repression of reporter mRNAs with 3′UTR of validated let-7a target genes TLR4 or HMGA2 after 4 h and 24 h of LPS treatment. (**F**) Repression of a miR-122 reporter in RAW 264.7 cells exogenously expressing miR-122 on LPS treatment. For all quantification of RT–PCR and luciferase data, mean values ± s.d. were determined from three independent experiments. LPS, lipopolysaccharide; miRNA, microRNA; mRNA, messenger RNA; RL, *Renilla* luciferase; rRNA, ribosomal RNA; RT–PCR, reverse transcriptase polymerase chain reaction; UTR, untranslated region.

**Fig 2 |** Loss of miRNA from Ago2 deactivates miRNPs in LPS-treated mammalian macrophages. (**A**) Ago2 binding to target mRNA in RAW 264.7 cells ▶ expressing a RL let-7a reporter and FLAG-HA-Ago2. Cells were lysed before and after 4 h of LPS treatment and from Ago2 IPed materials RNA was isolated and Ago2-associated RL-3xbulge-let-7a mRNA was quantified (left panel) (***$P<0.0002$). In similar but separate IP experiments, quantifications of Ago2-associated TNF-α, and IL-6 mRNAs were performed (right panel) (*$P<0.0376$). (**B**) A schematic representation of RL-5BoxB reporter used for Ago2-tethering experiments. In tethering assay, RL-5BoxB mRNA was co-expressed with either N-HA-Ago2 or HA-Ago2 and repression level of RL-5BoxB reporter was measured. Association of RL-5BoxB mRNA with N-HA-Ago2 in untreated or 4-h LPS-treated RAW 264.7 cells was quantified. (**C**) Relative level of active miR-122 RISC present in naïve and 4-h LPS-treated cells were quantified by *in vitro* RISC cleavage assays (cleaved product is marked by an *). For negative control, miRISC was prepared from naïve RAW 264.7 cells not expressing miR-122. A $^{32}$P 22 nt marker oligonucleotide was loaded as size marker (lane M). (**D,E**) Change in miRNA binding to Ago2 in LPS-activated macrophages. let-7a level in Ago2 IPed materials isolated before and after 4-h LPS treatment was quantified in RAW 264.7 or mouse primary macrophages isolated from peritoneal exudates (**D**) or in splenic macrophages of mice intraperitoneally injected with LPS or PBS (**E**). $n=3$ for LPS- or PBS-injected mice. (**F**) Levels of different miRNAs were quantified from HA-Ago2 immunoprecipitates obtained from naïve- or LPS-activated RAW 264.7 cells. (**G**) Let-7a miRNA association was measured from HA-Ago1 or HA-Ago3 immunoprecipitates before and after LPS treatment. For quantification purpose, mean values ± s.d. were determined from three independent assays. LPS, lipopolysaccharide; miRISC, miRNA-induced silencing complex; miRNA, microRNA; miRNP, microRNA protein complex; mRNA, messenger RNA; PBS, phosphate-buffered saline; RISC, RNA-induced silencing complex; RL, *Renilla* luciferase; TNF-α, tumour necrosis factor alpha.

Fig S1A online). Similarly, reporters containing the 3′UTR of let-7a target gene HMGA2 [13] or let-7a-regulated TLR4 [6] showed transient reduction in miRNA-mediated repression (Fig 1E). Additionally, a GFP-reporter with let-7a binding sites also showed derepression on LPS treatment (supplementary Fig S1B online). Derepression of miRNA activity was not exclusive for let-7a, as exogenous expression of liver-specific miR-122 in RAW 264.7 cells coexpressing a miR-122 reporter showed a similar trend (Fig 1F). Repression of a RL reporter with one perfect let-7a binding site in its 3′UTR was also relieved by LPS stimulation (supplementary Fig S1C online). Subsequently, we tested the activity of a small interfering RNA targeting the coding sequence of RL mRNA and observed a drop in small interfering RNA activity with LPS (supplementary Fig S1C online). A similar reduction in miRNA activity at early hours of LPS stimulation in human monocytic cell line THP1 was also observed (supplementary Fig S1D online).

We treated cells with phorbol 12-myristate 13-acetate or hypomethylated DNA (CpG) as other TLR4-independent activators of macrophage and documented comparable decrease of miRNA activity during initial period (supplementary Fig S1E online). LPS isolated from a different origin (*Salmonella typhimurium*) also showed a similar trend (supplementary Fig S1F online). These results indicate that the stimulation with different activators induces transient decrease of miRNA activity in mammalian macrophage cells. In Dicer1-depleted RAW 264.7 cells, TNF-α

mRNA level was augmented supporting a direct role of miRNAs in controlling the early phase of proinflammatory response in mammalian macrophages (supplementary Fig S1G online).

## No change in let-7a level in LPS-activated macrophage

Downregulation of miRNA activity could be either due to a decrease in miRNA level or degradation of specific miRNP components. We did not find any decrease in let-7a expression in RAW 264.7 or primary macrophage cells with LPS treatment (supplementary Fig S2A online). Level of exogenously expressed miR-122 was also unaltered with LPS stimulation (supplementary Fig S2A online). Expression of Ago2 or other important P-body factors did not change with LPS treatment (supplementary Fig S2B online). Of the four Argonautes, Ago2 is the most abundant in RAW 264.7 cells (supplementary Fig S2C online). RNA processing bodies or P-bodies can serve as the storage sites for miRNA-repressed messages in mammalian cells and under stress condition repressed mRNAs can be relocalized from P-bodies to translating polysomes in human hepatic cells [14]. Status of P-bodies (size and number) was unaltered with LPS stimulation (supplementary Fig S2D online), but we documented an impaired P-body localization of Ago2 in 4 h LPS-treated RAW 264.7 cells (supplementary Fig S2E,F online). Does Ago2 directly affect transcription of proinflammatory cytokine genes because of an increased nuclear localization in LPS-stimulated macrophages? We neither observed any change in nuclear localization

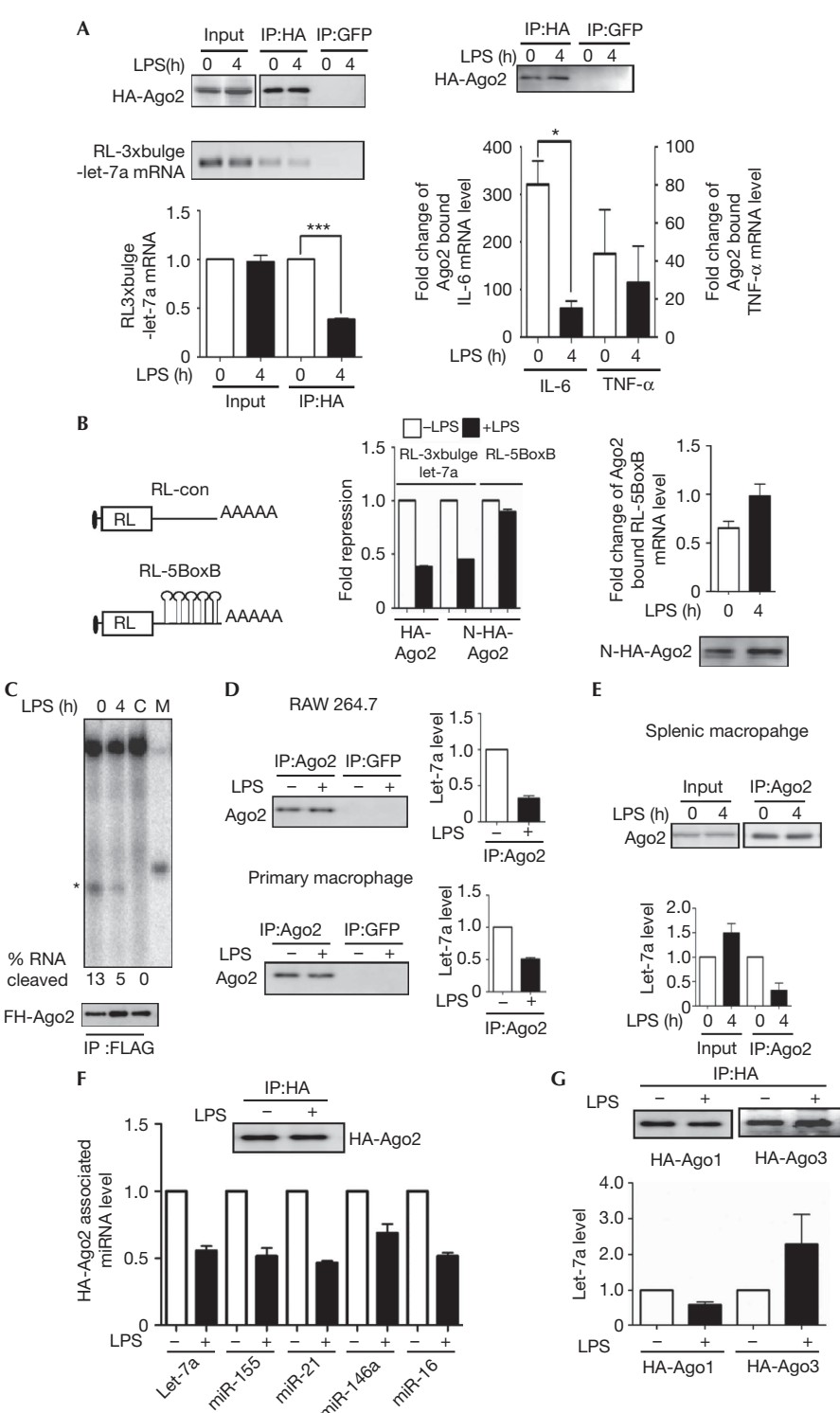

of Ago2 nor did it show any increased association with the promoter element of TNF-α on LPS exposure (supplementary Fig S2G online). In animal cells, multivesicular bodies (MVB) are important for miRNA-mediated repression and at subcellular level, MVBs are enriched with miRNAs [15]. In LPS-treated RAW 264.7 cells, miRNA showed a reduced association with

MVBs along with a subsequent increase of miRNA content in high-density fractions. (supplementary Fig S2H online).

**miRNA activity reversal is *de novo* transcription independent**
LPS treatment did not alter the RL reporter mRNA level whereas an elevated protein translation happened at early hours of LPS

treatment (supplementary Fig S3A,E online). Application of α-amanitin, that blocks synthesis of mRNA transcripts, had no effect on fold derepression of let-7a reporter (supplementary Fig S3B online). Cells were transfected with *in vitro* synthesized RL reporter mRNAs containing miR-122 binding sites and similar derepression was also observed with LPS stimulation (supplementary Fig S3C,D online). Thus, the derepression is independent of *de novo* transcription of the target RNA and reversal of miRNA-mediated repression primarily happens for pre-existing miRNA-repressed mRNAs. The reversal of repression with LPS treatment resulted in a shift of the let-7a reporter mRNA to polysomes suggesting active translation of the reporter mRNA in 4 h LPS-treated cells (supplementary Fig S3E online). When this translational upregulation was slowed down by rapamycin, derepression of the let-7a reporter was partially blocked (supplementary Fig S3F online). Similarly, a reporter with a 5′UTR of *p27* gene that slows down the translation of the *cis*-encoded mRNA due to presence of a translation inhibitory secondary structure was found to be defective to get derepressed (supplementary Fig S3G online).

## Loss of miRNA from Ago2 in LPS-treated macrophage

Evident from the experiments described above, impairment of miRNA activity could not be due to lowering of miRNA or Ago2 protein expression. However, LPS stimulation did alter the polysomal distribution of let-7a but not of Ago2 protein in RAW 264.7 cells (Supplementary Fig S3E online). From this experiment, it seems that there was a loss of miRNA from polysome-associated Ago2 in LPS-treated cells. We did Ago2 immunoprecipitation (IP) and observed a reduced association of target mRNAs at the early phase of LPS stimulation (Fig 2A). Tethering of hAgo2 to mRNA inhibits the protein translation from the hAgo2-tethered message [12]. Interestingly, the target RNA binding and repression achieved by Ago2 tethering could not be reversed during the LPS stimulation (Fig 2B). Ago2 tethering-mediated repression is independent of miRNA–mRNA base pairing and therefore this observation prompted us to speculate that the miRNA binding to Ago2 might get affected during LPS-mediated activation of macrophage. In an *in vitro* RISC cleavage assay, performed with miRISC isolated from naïve- and LPS-treated macrophage, differences in Ago2 slicer activity was documented (Fig 2C). The impaired slicing function was consistent with low miRNA binding of Ago2 in LPS-treated RAW 264.7, mouse primary macrophages and also in THP1 cells (Fig 2D and supplementary Fig S4A online). Low let-7a binding of Ago2 in macrophage also happened *in vivo* after treating Balb/C mice with LPS (Fig 2E). Not only let-7a, other miRNAs also dissociated from Ago2 with LPS stimulation (Fig 2F and supplementary Fig S4B online). We tested miRNA association with other Ago proteins. In LPS-activated RAW 264.7 cells, let-7a association with HA-Ago1 was decreased but it increased with HA-Ago3 (Fig 2G). Interestingly, association of Ago2 with GW182 protein also gets affected in activated macrophage (supplementary Fig S4C online), which might explain the reduction of Ago2 localization in P-body in LPS-activated cells (supplementary Fig S2D–F online).

## Ago2 phosphorylation reduces miRNA binding in macrophage

Previously, it has been shown that phosphorylation of hAgo2 at Tyr-529 interferes with miRNA binding [16]. We did IP with a phospho-tyrosine-specific antibody and detected Ago2 level in the IP material. With LPS stimulation, the amount of IPed Ago2 protein was increased at 4 h (Fig 3A). Reciprocally, with IPed Ago2 from LPS-activated RAW 264.7 cells or primary macrophage; we also found an increase in Tyr phosphorylation (Fig 3A and supplementary Fig S4D online). Similar increase in phosphorylation of exogenously expressed HA-Ago2 was also documented on LPS treatment (supplementary Fig S4D online). The mutant version of the Ago2 protein (FH-Ago2Y529F) that cannot be phosphorylated at Tyr at 529 position remained bound to miRNA and was insensitive to LPS treatment (Fig 3B). We isolated FH-Ago2 or Y529F mutant from HEK293 cells co-expressed with miR-122. Isolated miR-122 miRISC was treated with naïve- or LPS-activated RAW 264.7 cell extract. Incubation with LPS-treated extract showed a reduction in bound miRNA (supplementary Fig S4E online). This reduction was also observed in an *in vitro* RISC cleavage activity accompanied by an increased phosphorylation of FH-Ago2 containing miRNP but not for the Y529F mutant in LPS-treated macrophage (Fig 3C,D). Expression of FH-Ago2Y529F in endogenous Ago2-depleted cells resulted in defective derepression of miRNA-targeted messages in RAW 264.7 cells whereas repression in cells expressing a phosphomimetic mutant of Ago2, FH-Ago2Y529E, remained unaffected with LPS stimulation (Fig 3E).

Importantly, the dissociation of miRNA from HA-Ago2 and increase in Y529 phosphorylation were partially blocked in the presence of p38 mitogen-activated protein kinases (MAPK) inhibitor, SB 203580. Derepression of miRNA activity was also partially blocked in the presence of p38 MAPK inhibitor but not with inhibitors specific to ERK and JNK pathways (supplementary Fig S4F online). So, p38 MAPK pathway is involved in LPS-driven phosphorylation of Ago2 and concomitant loss of miRNP activity in macrophage cells.

## miRNP unloading controls inflammatory response

Macrophages respond to external stimuli by inducing the expression of proinflammatory cytokines. Endogenous Ago2 was depleted with siAgo2 against 3′UTR of Ago2. FH-Ago2 or FH-Ago2Y529F, resistant to siAgo2, was expressed in those cells and different cytokine levels were measured with LPS stimulation. In response to LPS, the FH-Ago2Y529F-expressing cells were defective to elevate the expression of proinflammatory cytokines (Fig 4A).

Infection with invading pathogens is prevented by proinflammatory cytokines expressed in host macrophage. Thus, macrophage cells primed with low doses of LPS to express proinflammatory cytokines show resistance to infection by *L. donovani,* a protozoan parasite that infects and induces anti-inflammatory response in macrophages [8]. Parasitic infection of the macrophage switches the proinflammatory response to an anti-inflammatory one with concomitant increase in expression of IL-10 [17]. The observed increase in IL-10 and decrease in TNF-α or IL-6 mRNA levels in Y529F mutant but not in the wild type of Y529E mutant expressing cells were indicative of elevated parasite infection (Fig 4B,C). LPS-primed RAW 264.7 cells, depleted for endogenous Ago2 and expressing FH-Ago2Y529F mutant, also showed increased internalization of *L. donovani* (Fig 4D,E). Thus, the Ago2Y529F ensured a reduced pro- but robust anti-inflammatory response in parasite-infected host cells. Defect in phosphorylation of Ago2 miRNPs during early infection phase possibly acts to prevent the loss of miRNA from Ago2 and reduce production of proinflammatory

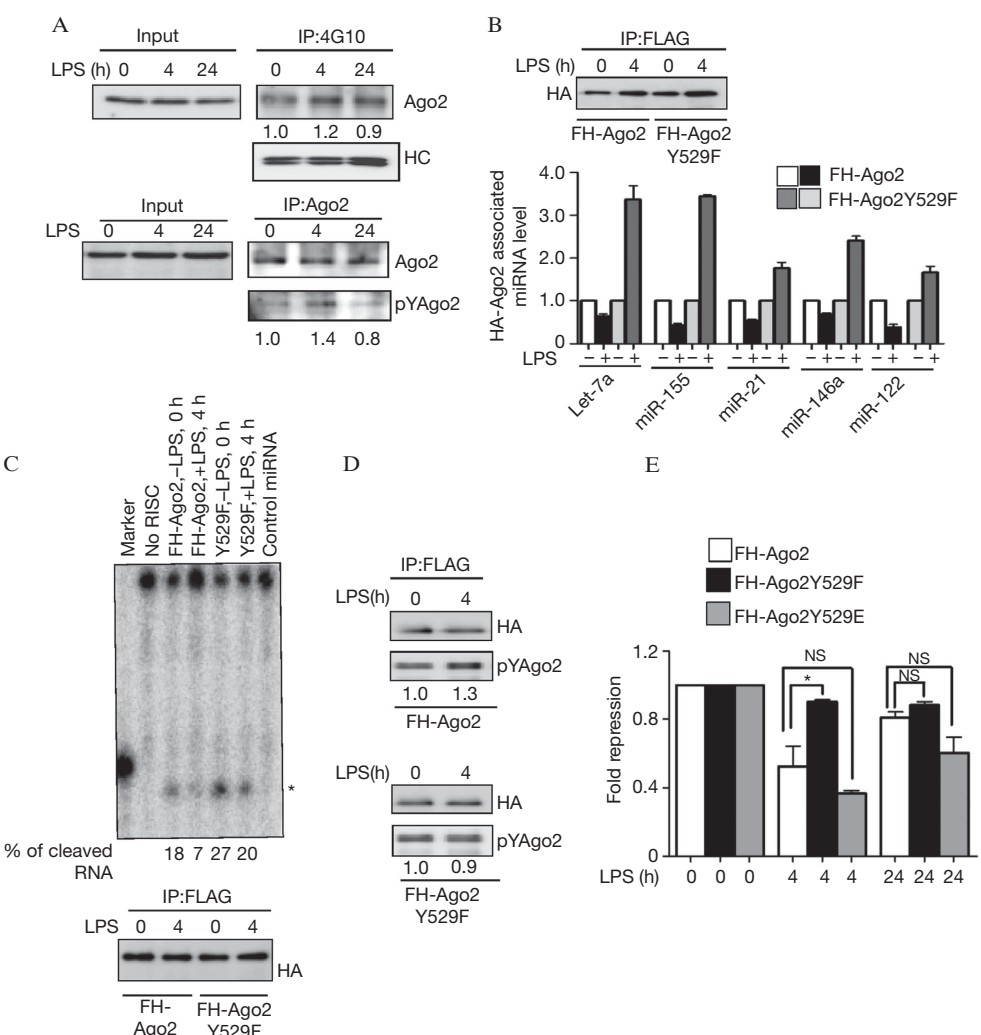

**Fig 3 | LPS-induced phosphorylation at Tyr-529 dissociates miRNA from Ago2 and causes downregulation of miRNA activity in RAW 264.7 cells.**
(A) Tyr phosphorylated proteins were IP(ed) with anti-phosphotyrosine antibody (4G10) from naïve and LPS-stimulated RAW 264.7 cell extracts and immunoblotted for Ago2 (upper panel). Reciprocally, endogenous Ago2 was IPed and immunoblotted for Tyrosine phosphorylated Ago2 (pYAgo2) using 4G10 antibody (lower panel). (B) miRNA binding to Ago2 and Y529F mutant. FH-Ago2 or FH-Ago2Y529F were IP (ed) from LPS-treated or naïve RAW 264.7 cells expressing miR-122 and FH-Ago2 or FH-Ago2Y529F-associated miRNA levels were quantified. (C–E) Ago2 phosphorylation and miRNA activity in RAW 264.7 cells. Purified miRISC isolated from naïve or LPS-activated RAW 264.7 cells coexpressing miR-122 and either FH-Ago2 or FH-Ago2Y529F was used for *in vitro* RISC cleavage assay. Cleaved product is marked by an * (C; upper panel). Ago2 level in isolated RISC was quantified by western blot (C; bottom panel). Phosphotyrosine status of isolated FH-Ago2 or FH-Ago2Y529F, before and after LPS activation, were detected using 4G10 antibody (D). Repression level of RL-3xbulge-miR-122 reporter in LPS-treated RAW 264.7 cells expressing miR-122 and coexpressing either FH-Ago2 or its mutants. Repression levels in LPS-untreated cells were used for normalization (*$P < 0.0450$) (E). For panels A and D, western blots were quantified and normalized against the signal intensities of the corresponding heavy chain of anti-HA antibody used for immunoprecipitation. For quantification purpose, mean values ± s.d. were determined from three independent experiments. HC, heavy chain; IP, immunoprecipitation; LPS, lipopolysaccharide; miRNA, microRNA; RISC, RNA-induced silencing complex; RL, *Renilla* luciferase.

cytokines. This might subsequently reduce miRNP recycling and prevent *de novo* miRNP formation. Interestingly, *L. donovani* cleaves Dicer1 to reduce new miRNA production in liver cells during the late phase of parasite invasion to downregulate miR-122 in liver and to reduce serum cholesterol [18]. Therefore, whether *L. donovani* can ensure a reduced *de novo* miRNP formation both by reducing miRNA production (as shown in Ghosh *et al* [18])

and also by preventing *de novo* miRNP formation by preventing miRNP recycling in infected macrophage cells is needed to be explored further.

Unlike that with let-7a, reversal of miRNA derepression after 24 h of LPS treatment of RAW 264.7 cells transfected with small interfering RNA or exogenous miR-122 was incomplete and only a fraction of initial activity was regained. This suggests

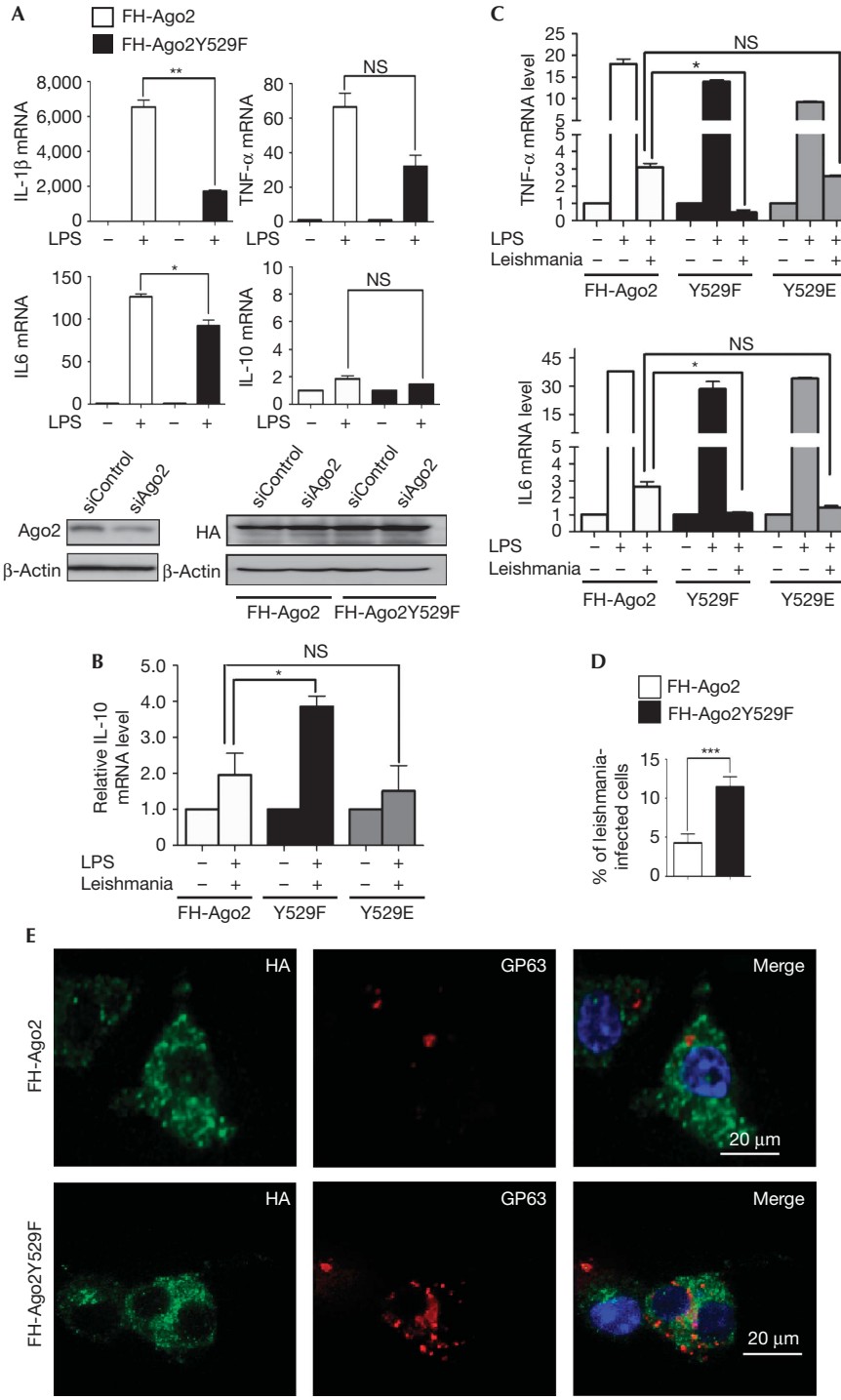

**Fig 4 | Transient decrease in miRNA activity is necessary for regulated expression of proinflammatory cytokines in LPS-activated RAW 264.7 cells.**
(**A**) FH-Ago2 or Y529F mutant were expressed in endogenous Ago2 knocked-down RAW 264.7 cells evident by Ago2 western blot. Levels of TNF-α, IL-1β, IL-6 or IL-10 mRNA were estimated in naïve and activated cells (**$P < 0.0069$ and *$P < 0.0444$). (**B–E**) Invasion of *L. donovani* in LPS-primed RAW 264.7 cells expressing FH-Ago2 or its mutant forms (Y529F or Y529E). RAW 264.7 cells depleted for endogenous Ago2 were pre-treated with LPS for 4 h followed by infection for 24 h with *L. donovani* and IL-10 mRNA level (**B**) (*$P < 0.0450$) or TNF-α and IL-6 mRNA level (**C**) (*$P < 0.0102$ and *$P < 0.0337$, respectively) were measured after infection. Parasite internalization was detected by indirect immunofluorescence using an antibody-specific for gp63, a Leishmanial membrane metalloprotease. Quantification was done by analysing more than 50 transfected cells (***$P < 0.0009$) (**D,E**). For quantification purpose, mean values ± s.d. were determined from three independent assays. IL, interleukin; LPS, lipopolysaccharide; miRNA, microRNA; mRNA, messenger RNA; TNF-α, tumor necrosis factor alpha.

reloading of pre-existing miRNAs to form miRISC in activated macrophage is partial and might be preferential for new miRNAs synthesized. Reversible miRNA unloading of Ago2 allows the reshuffling of miRNPs in activated macrophage when newly synthesized miRNAs would win over the pre-existing or used miRNAs to form miRNPs.

We have found that, the Tyr phosphorylation status of Ago3 was decreased (supplementary Fig S4G online) and the association of let-7a miRNA with Ago3 was increased with LPS stimulation (Fig 2G). hAgo1 (NP_036331) and hAgo3 (NP_079128) proteins show very high structural similarity with respect to hAgo2 structure (supplementary Fig S5A online; root mean square deviation: 0.3 Å in all cases). However, significant differences in total and accessible volume of the largest cavity, binding/unbinding energies and types of interacting residues with respect to ATP were observed between hAgo2 and hAgo3 proteins (supplementary Fig S5B–D online and supplementary Movies 1 and 2 online). These differences could suggest a more preferred binding of ATP and Y529 phosphorylation of hAgo2 than that of hAgo3.

Phosphorylation at S387 has been suggested to be a key determinant for increased activity of miRNPs in mammalian cells [19]. LPS stimulation did not increase phosphorylation of Ser residues of Ago2 (supplementary Fig S4H online). S387 residue is located far apart (supplementary Fig S5E online) from the Y529 (49 Å) and the bound miRNA (30 Å) that suggests a minimal or no direct role of phosphorylation of S387 on Y529 phosphorylation and miRNA binding.

Taken together, our findings have revealed how the initial flux of proinflammatory cytokine expression is achieved by uncoupling of miRNA from Ago2 protein followed by a restoration of miRNA-mediated repression possibly to control hyper-responsiveness. Investigating further details will facilitate a better understanding of the mechanism, identification of the kinase responsible for Ago2 phosphorylation and also of help in identifying new candidates to control inflammation.

## METHODS

**Cell culture and reagents.** RAW 264.7and THP1 were cultured in RPMI 1640 medium (Gibco) supplemented with 2 mM L-glutamine, 0.5% β-mercaptoethanol and 10% heat-inactivated fetal calf serum. The macrophage cells were stimulated with 1 ng/ml *Escherichia coli* O111:B4 LPS (Calbiochem, La Jolla, CA).

**Immunoprecipitation.** For IP reactions, cells were lysed by sonication and clarified by centrifugation. Clear supernatants were then incubated either with antibody pre-bound Protein G Agarose bead (Invitrogen) or pre-blocked anti-FLAG M2 affinity gel (Sigma; A2220) and rotated at 4 °C for 4 h. Subsequently, the beads were washed thrice and the bound proteins were analysed by western blotting. From half of the bound beads, separated during final washing steps, RNA was extracted using TRIzol LS (Invitrogen) and used for real-time polymerase chain reaction (PCR) of miRNA and mRNA.

**RNA isolation and miRNA or mRNA detection.** Total RNA was extracted by using the TRIzol reagent (Invitrogen) and Northern Blotted for miRNAs. For quantification of different miRNAs, TaqMan Universal PCR Master Mix (Applied Biosystems) was used following the manufacturer's instructions. For mRNA quantification, total RNA was used to prepare cDNA with random nonamers (Eurogentec Reverse Transcriptase Core Kit) and produced cDNA

was used for PCR amplification with gene-specific primers with MESA GREEN qPCR Master Mix Plus (Eurogentec). Procedures are detailed in supplementary information online.

**Statistical analysis.** All graphs and statistical analyses were generated in GraphPad Prism 5.00 (GraphPad, San Diego, CA). Nonparametric unpaired *t*-test or paired *t*-test was used for analysis. *P*-values $<0.05$ were considered to be statistically significant and $>0.05$ were not significant (NS). Error bars indicate mean ± s.d. Results of the statistical analysis are mentioned in the respective figure legends.

ACKNOWLEDGEMENTS
We would like to thank Witold Fillipowicz and Gunter Meister for different plasmid constructs. We also thank Syamal Roy for discussion; Bahnisikha Barman and Kshudiram Naskar for helping us with experimental works. We thank the funding bodies, Wellcome^Trust, UK (084324/Z/07/A) and CSIR (Council of Scientific and Industrial Research), Government of India (BSC 0114). A.M., M.B. and A.C. are recipients of CSIR fellowships.

*Author contributions*: A.M. and S.N.B. designed the experiments and analysed the data. A.M. performed all of the experimental work, except the molecular dynamics-related work (performed and analysed by A.C. and S.C.) and *in vitro* RISC cleavage assay (done by M.B.). S.N.B. has conceived and managed the project. All authors contributed to manuscript preparation.

CONFLICT OF INTEREST
The authors declare that they have no conflict of interest.

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
