## [Review Process File · EMBO Reports]

Manuscript EMBOR-2013-37763

A transient reversal of miRNA-mediated repression controls macrophage activation

Anup Mazumder, Mainak Bose, Abhijit Chakraborty, Saikat Chakrabarti and Suvendra N. Bhattacharyya

Corresponding author: Suvendra N. Bhattacharyya, CSIR-Indian Institute of Chemical Biology

Review timeline:

Submission date:	12 November 2012
Editorial Decision:	19 December 2012
Revision received:	31 March 2013
Editorial Decision:	19 April 2013
Resubmission received:	15 July 2013
Editorial Decision:	06 August 2013
Revision received:	19 August 2013
Accepted:	22 August 2013

Editor: Alejandra Clark, Nonia Pariente

Transaction Report:

1st Editorial Decision

19 December 2012

Thank you for the submission of your research manuscript to EMBO reports. We have now received the full set of reports on it, which I copy below.

As you will see, although all referees agree that the findings are potentially interesting and novel, they raise a number of significant concerns that need to be experimentally addressed to make the study fully conclusive. Both referees #1 and #2 indicate that the proposed model of a transient release of miRNA-mediated repression upon macrophage activation, is not fully supported by the data. Referee #2 is concerned by the transcriptional activation of miRNA target genes upon LPS treatment, the use of Ago2 slicing activity as a readout of miRNA binding and the fact that other Ago proteins should be examined for their effects upon macrophage activation. Referee #1 suggests that the transient decrease in let-7 activity upon LPS treatment might be an indirect effect of LPS-induced miRNAs and their displacement of let-7 from the RISC complex. In addition, referee #3 suggests further experiments to assay whether the transient reversal of miRNA-mediated repression is observed for a few or many miRNAs. Furthermore, all referees point to various technical issues including additional controls, quantification and statistical analysis of the data.

Given the potential interest of the novel findings and considering that all referees provide constructive suggestions on how to move the study forward, I would like to give you the opportunity to revise the manuscript, with the understanding that the major referee concerns, including those mentioned above, have to be addressed and that acceptance of the manuscript would entail a second round of review. I would like to point out that it is EMBO reports policy to allow a single round of revision only, and that thus acceptance or rejection of the manuscript will depend on the outcome of the next final round of peer-review.

Revised manuscripts should be submitted within three months of a request for revision; they will otherwise be treated as new submissions. If you feel that this period is insufficient to address the referees' concerns I can potentially extend this period slightly. Also, the length of the revised manuscript should not exceed roughly 30,000 characters (including spaces). Should you find the length constraints to be a problem, you may consider including some peripheral data in the form of Supplementary information. However, materials and methods essential for the understanding of the key experiments should be described in the main body of the text and may not be displayed as supplemental information only.

We have also started encouraging authors to submit the raw data for western blots (i.e. original scans) to our editorial office. These data will be published online as part of the supplementary information. This is voluntary at the moment, but if you agree that this would be useful for readers I would like to invite you to supply these files when submitting the revised version of your study.

As part of the EMBO publication's Transparent Editorial Process, EMBO reports publishes online a Review Process File to accompany accepted manuscripts. This File will be published in conjunction with your paper and will include the referee reports, your point-by-point response and all pertinent correspondence relating to the manuscript.

We also welcome the submission of cover suggestions or motifs that might be used by our Graphics Illustrator in designing a cover.

I look forward to seeing a revised form of your manuscript when it is ready. Should you in the meantime have any questions, please do not hesitate to contact me.

REFeree REPORTS:

Referee #1

The manuscript by Mazumder et al. explores the molecular mechanism of cytokine gene regulation by miRNAs during an inflammatory response. The authors found that expression of several proinflammatory cytokine genes that are subject to miRNA control in macrophages is transiently upregulated despite the fact that the level of expression of the miRNAs that target these cytokine genes was not significantly changed. They concluded that this transient relief of miRNA repression in macrophages is due to inflammation-induced phosphorylation of the AGO2 protein, a key component of the RISC machinery. Transient phosphorylation of AGO2 at Tyr 529 position inhibits AGO2 binding to miRNAs that negatively regulate cytokine expression and thus, cells expressing AGO2 Tyr 529 mutants are less sensitive to proinflammatory stimuli.

While I find these findings quite novel and very intriguing, I feel that the authors failed to conclusively rule out some of the alternative explanations of their results (see major reservations below) and therefore the manuscript in its current form should not be published in EMBO Reports.

Major reservations:

1. The transient decrease in let-7a miRNA activity in LPS treated macrophages can also occur as a result of a dramatic increase in expression of LPS-induced miRNAs (miR-155, miR-21, miR-146, etc.). Assuming the number of AGO2/RISC complexes stays constant upon LPS treatment (as the authors suggest in Fig.S2B), these newly synthesized inducible miRNA molecules should displace some of the steady-state miRNAs (like let-7a) from the RISC. As a result, a decrease in let-7a activity will be observed. The authors should account for such possibility. They should analyze the levels of the LPS-induced miRNAs in AGO2 immunoprecipitates from stimulated macrophages and correlate these levels with the levels of let-7a bound to AGO2. Along the same line, they should define how phosphorylation of Tyr 529 in AGO2 affects loading and activity of the LPS-inducible miRNAs. It is totally plausible that AGO2 Tyr529 mutants have a decreased binding capacity for the LPS-inducible miRNAs and therefore some of the functional defects that were observed by the authors in macrophages expressing AGO2Y529F mutant are due to a decrease in activity of these inducible miRNAs.
2. There are some disconnects between experimental data that put the authors' main hypothesis under question. For example, the amount of AGO2-bound IL-6 mRNA decreases significantly upon LPS treatment (Fig. 2A), while the level of AGO2-bound TNF mRNA was not altered by the same treatment (Fig.2A). However, the introduction of the AGO2 Tyr529 phosphorylation mutant into macrophages seems to affect the TNF mRNA induction to a larger extent than the IL-6 induction (Fig.4A). How do the authors reconcile these results? Perhaps, they should examine the levels of AGO2-bound IL-6 and TNF mRNAs in AGO2 immunoprecipitates from cells that express AGO2Y529F mutant to help resolve the discrepancy.

Minor reservations:

1. To confirm the specificity of let-7a-mediated derepression of its target genes upon LPS treatment, I suggest the authors repeat experiment described in Fig. 1D using reporter construct with mutated let-7a bulge sites. Alternatively, they can perform experiment described in Fig. 1E in the presence of let-7a mimic.
2. Some of the described in the manuscript effects are rather small. Thus, it would be helpful to have quantification of most gel results. For example, what is the increase in the level of AGO2 phosphorylation in Fig. 3A and D?
3. Some experiments are poorly described in the text and some figure legends are missing proper explanation of the experimental design. For example, results of experiments described in Fig. 1B and C are poorly explained in the text. The legend for the experiment described in Fig. 3E does not specify how the normalization of the reporter assay results was done. Were all results here normalized to untreated FH-AGO2 level?
4. Statistical analysis of some of the data is missing. For example, the P value for experiments described in Fig. 4A should be calculated.
5. Ago2 phosphorylation in response to LPS seems to be transient in Fig.3A (lower panel), but fails to go back to basal levels in the upper panel. Why is such a difference between results? This should be discussed.
6. The authors have a tendency to overstate the facts. For example, in the last paragraph on page 7 they state: "When this translational upregulation was partially blocked by application of rapamycin, derepression of the let-7a reporter was prevented (supplementary Fig S3D online)." According to my analysis of the figure, the derepression of let-7a reporter was partially blocked, but not prevented.
7. Use different colors or patterns in graphs in Fig. S1C, E and G. It is hard to differentiate between the two datasets.

Referee #2

In this manuscript, Mazumder and collaborators demonstrate quite convincingly that the phosphorylation the tyrosine 529 of Ago2 occurred upon LPS treatment and the Argonaute localization is affected by this treatment. Unfortunately, the authors do not convincingly demonstrate the release of miRNA-mediated repression in LPS treated cells. Prior considering this interesting study for publication in EMBO Reports, the following experiments and controls will be required to better support their conclusions:

- 1- If I understand correctly the authors' model, it is proposed that upon infection, the miRNA

repression will be released to support cytokine expression. In Figure 1, it is clear that the expression level of the IL-6 mRNA is affected upon treatment with LPS. From this set of data, one can argue that the activation of macrophages by LPS does not relieve the miRNA repression but instead increases the miRNA targets transcription leading to an increase of their expression levels. It will be important to convincingly support the authors model to demonstrate that the transcription of endogenous miRNA targets does not solely contribute to the increase of protein levels. From the data presented in Figure S3, it appears that the level of miRNA reporters is not affected upon LPS and alpha-amanitin treatment but knowing that these constructs are under control of strong promoters, they are not optimal to rule out any effect on transcription. To measure the effect on transcription, the authors should measure the level of both pre-mRNA and mRNA molecules.

2- I am not convinced that measuring the slicing activity is a reliable readout of the miRNA binding to Ago2. Upon LPS treatment, post-translational modifications of Ago2 can affect independently its capacity to bind small RNAs and its slicing activity. For this reason, it is suitable to precisely measure instead the amount of miRNAs associated to Argonaute complexes. The authors presented immunoprecipitation data that indicate that the level of miRNA found in the Ago2 complex decrease upon LPS treatment but because others human Agos can also contribute to the miRNA-targeted repression, the interaction of all four human Agos with miRNAs upon LPS treatment should be assessed as well. To circumvent the lack of reliable Argonaute specific antibodies for IP, the authors should perform miRNA pulldown complexes using 2'-O-methylated antisense RNA column with more than one miRNA (at least with one more (miR-122 or miR-16) along with let-7a). In all cases, it will be important to measure the levels of miRNAs in the input fractions to confirm that their levels are not affected upon LPS treatment.

3- As mentioned before, it is known that all four Ago proteins can cause miRNA-specific translational repression. I therefore do not understand why knocking down Ago2 and supplementing cells with AGO2Y529F abrogate the miRNA-targeted repression. It was previously reported that this specific phosphorylation site is also found in other human Argonautes (Rudel et al, NAR, 2011). What are the levels of Ago1, 3 and 4 proteins in macrophages and RAW 264.7 cells? Are they expressed at the same level of Ago2? These important informations should be provided and discussed by the authors.

4-Figure 1C and S3C: Because the Northern blots associated to the fractionation experiments show different band intensity, it is difficult to conclude that the association of IL-6 and reporter mRNAs with polysomal fractions is decreased upon cell treatment with LPS. Proper quantification will be required.

5-Figure 4: It is important to show the Ago2 siRNA knockdown does not affect FH-AGO2 and FH-AGO2Y529F expression. Western blots should be provided.

6-Figure S4: To support that the interaction of Ago2 with GW182 is also affected upon LPS treatment, quantification should be provided.

7-Figure 2B: Is the level of repression obtained with the BoxB system comparable with the repression detected with the reporter carrying the let-7a sites? If is not the case, the way the data are currently presented can be misleading.

Referee #3

The manuscript by Mazumder et al provides evidence that microRNA repression of pro-inflammatory cytokine mRNAs is transiently attenuated during the early stages of the macrophage inflammatory response. Consequently, cytokine production is enhanced. The authors also claim that this process is not due to altered miRNA expression levels, but rather a result of decreased Ago2 association with miRNAs. Motivated by a previous report, the study points to phosphorylation of Ago2 at Tyrosine 529 as a critical switch that decreases miRNA binding. It is also demonstrated that macrophages expressing a loss of function Y529 mutant Ago2 produce less cytokines and have a decreased ability to kill *Leishmania* in vitro.

While the overall concepts in this paper are interesting, there are some significant issues that need to be addressed with new data to support the claims. First, the proposed mechanism is not overly novel because phosphorylation of Tyrosine 529, as the authors acknowledge, has previously been shown to diminish miRNA binding. While the current study does provide a biological context for this, it would be important to know which kinase, or at least which signaling pathways, are being activated during inflammation and subsequently mediate phosphorylation of Ago2 at Y529. The other issue is the lack of data for one to be able to grasp the breadth of this suppression of the miRNA pathway shortly after macrophage activation. Does it only impact a few miRNAs, or miRNAs in general? This could be addressed using HITS-CLIP before and after LPS treatment to determine which miRNAs and specific mRNA targets are impacted by the proposed mechanism.

Other issues:

1. The authors include too much data to simply demonstrate the well-known fact that macrophages make inflammatory cytokines when they are activated by PAMPs.
2. More work should be done in primary macrophages instead, or in addition to, cell lines to confirm findings in physiologically relevant cell types.
3. In addition to experiments with loss of function mutations at Y529, the authors should also test the effects of a phosphomimetic at Y529, which should constitutively inhibit miRNA-mediated repression and boost cytokine production.
4. During experiments where mutant Ago2 is added to Raw264.7 cells with knocked down endogenous Ago2, how efficient was the transfection of mutant Ago2 constructs? What percentage of the cells are receiving the expression constructs? Raw264.7 cells can be challenging to transfect and the authors should make this determination. Based upon the observed impact on cytokine production, one would have to achieve a fairly high level of transfection.
5. Is it possible that phosphorylation of Ago2 at Y529 can impact cytokine expression through a mechanism that is independent of the miRNA pathway? For example, there have been some reports of Ago proteins associating with gene promoters in the nucleus. OR, Ago2 might be part of a signaling cascade downstream from TLR signaling.
6. It would also be useful to knockdown Dicer and see if enhanced cytokine production is observed. This would further support a role for miRNAs in suppressing the initial burst of cytokine production by macrophages.
7. There are several typos and grammatical mistakes throughout the manuscript that should be corrected.

1st Revision - authors' response

31 March 2013

Point-by-point response to the Referees' comments

Referee # 1:

Major comments:

1. *The transient decrease in let-7a miRNA activity in LPS treated macrophages can also occur as a result of a dramatic increase in expression of LPS-induced miRNAs (miR-155, miR-21, miR-146, etc.). Assuming the number of AGO2/RISC complexes stays constant upon LPS treatment (as the authors suggest in*

Fig.S2B), these newly synthesized inducible miRNA molecules should displace some of the steady-state miRNAs (like let-7a) from the RISC. As a result, a decrease in let-7a activity will be observed. The authors should account for such possibility. They should analyze the levels of the LPS-induced miRNAs in AGO2 immunoprecipitates from stimulated macrophages and correlate these levels with the levels of let-7a bound to AGO2. Along the same line, they should define how phosphorylation of Tyr 529 in AGO2 affects loading and activity of the LPS-inducible miRNAs. It is totally plausible that AGO2 Tyr529 mutants have a decreased binding capacity for the LPS-inducible miRNAs and therefore some of the functional defects that were observed by the authors in macrophages expressing AGO2Y529F mutant are due to a decrease in activity of these inducible miRNAs.

We appreciate the comment that prompted us to test the possible alternative explanation of our results. We have measured level of LPS-induced miRNAs in HA-AGO2 immunoprecipitated materials isolated both from naïve and LPS activated RAW 264.7 cells and documented similar reductions in Ago2 bound miRNA levels for all the miRNAs tested (Fig. 2F). Therefore loss of miRNAs from Ago2 in LPS-treated macrophage seems to be a universal event that happens irrespective of abundance of a miRNA. The notion was confirmed further in another experiment where we overexpressed miR-122 exogenously in RAW 264.7 cells and measured miR-122, let-7a or LPS-induced miRNAs from Ago2 immunoprecipitated materials (Supplementary Figure S4C online). We failed to detect a different trend for any of the miRNAs tested and all of them showed a reduced Ago2 binding after LPS treatment of 4h even in a context where let-7a miRNPs were purposefully outnumbered by miRNP-122. Therefore the hypothesis of replacement of a miRNA from existing miRNPs with another abundant miRNA causing the derepression of pre-existing miRNP repressed messages seems to be non-operative in LPS-activated macrophage cells. Thus reduction in miRNA activity in LPS-treated cells caused by a global loss of miRNAs from Ago2 seems

to be responsible for the observed effect on let-7a activity downregulation in LPS stimulated macrophages. As displacement seems to be non-operative in LPS activated macrophage during initial hours of activation, the effect of AGO2Y529F expression could not be due to defective binding of inducible miRNAs to mutant AGO2 in LPS treated cells.

Additionally, it may be noted that derepression of let-7a activity is transient and at 24h even when the inducible miRNAs are abundant and let-7a does not show a significant increase in its expression, repressive activity of let-7a regained back to initial level! This also supports the Ago2-phosphorylation dependent reversible unloading of let-7a rather a mere replacement of it from Ago2 by inducible miRNAs.

2. There are some disconnects between experimental data that put the authors' main hypothesis under question. For example, the amount of AGO2-bound IL-6 mRNA decreases significantly upon LPS treatment (Fig. 2A), while the level of AGO2-bound TNF mRNA was not altered by the same treatment (Fig.2A). However, the introduction of the AGO2 Tyr529 phosphorylation mutant into macrophages seems to affect the TNF mRNA induction to a larger extent than the IL-6 induction (Fig.4A). How do the authors reconcile these results? Perhaps, they should examine the levels of AGO2-bound IL-6 and TNF mRNAs in AGO2 immunoprecipitates from cells that express AGO2Y529F mutant to help resolve the discrepancy.

This is an interesting aspect. We measured the amount of IL-6 and TNF- α mRNAs associated with FH-AGO2 and FH-AGO2Y529F in RAW 264.7 cells and have documented almost four fold increased binding of both mRNAs to the phosphorylation defective mutant of AGO2 (Figure shown below). But from this data we could not conclude the possible reason of differences in results described in Fig. 2A and Fig. 4A.

Binding of TNF- α to the phosphomimetic mutant of AGO2 (FH-AGO2Y529E), that can't bind miRNA (Rudel S. *et.al*, Nucleic Acid Res, 2010), is

insensitive to LPS (as shown below). This is specific for TNF- α mRNA and not true for the GAPDH mRNA (used as control). Similar miRNA independent binding of FH-AGO2 may also contribute to the observed reduced effect of LPS on its binding to TNF- α mRNA. Additionally, unlike IL-6, TNF- α mRNA has AU rich elements (ARE) and its stability is regulated by other ARE binding proteins like AUF1, TTP and HuR that are also regulated in activated macrophage (McMulle MR. *et. al*, JBC, 2003). Therefore the effect of LPS on TNF- α is a cumulative effect of these factors binding to 3'UTR of TNF- α mRNA and that may account for the differences.

Minor reservations:

1. To confirm the specificity of let-7a-mediated derepression of its target genes upon LPS treatment, I suggest the authors repeat experiment described in Fig. 1D using reporter construct with mutated let-7a bulge sites. Alternatively, they can perform experiment described in Fig. 1E in the presence of let-7a mimic.

We have performed the luciferase assays using reporters with mutated let-7a bulge sites and included them in Supplementary Figure S1A. We have also performed the luciferase assay of RL HMGA2UTR reporter in the presence of let-7a mimic after LPS stimulation and have incorporated it also in Supplementary

Figure S1A. In both cases similar let-7a activity derepression were observed with LPS.

2. Some of the described in the manuscript effects are rather small. Thus, it would be helpful to have quantification of most gel results. For example, what is the increase in the level of AGO2 phosphorylation in Fig. 3A and D?

Quantification are added to the corresponding western blots related to Ago2 and phosphorylated Ago2.

3. Some experiments are poorly described in the text and some figure legends are missing proper explanation of the experimental design. For example, results of experiments described in Fig. 1B and C are poorly explained in the text. The legend for the experiment described in Fig. 3E does not specify how the normalization of the reporter assay results was done. Were all results here normalized to untreated FH-AGO2 level?

All experiments including the ones described in Fig. 1B and C are now explained properly in the result section. Normalizations of results are also described in the figure legend.

4. Statistical analysis of some of the data is missing. For example, the P value for experiments described in Fig. 4A should be calculated.

The P values are now shown in Fig. 4A.

5. Ago2 phosphorylation in response to LPS seems to be transient in Fig.3A (lower panel), but fails to go back to basal levels in the upper panel. Why is such a difference between results? This should be discussed.

We addressed the comment. In case of immunoprecipitation using anti-phosphotyrosine antibody (upper panel), the immunoprecipitated material of 24 hr time point was little higher compared to 0 and 4 hr time points as evident from heavy chain western blot data included. The normalized quantification data performed against the immunoprecipitated heavy chain are now shown in Fig 3A and is consistent with the expected results.

6. *The authors have a tendency to overstate the facts. For example, in the last paragraph on page 7 they state: "When this translational upregulation was partially blocked by application of rapamycin, derepression of the let-7a reporter was prevented (supplementary Fig S3D online)." According to my analysis of the figure, the derepression of let-7a reporter was partially blocked, but not prevented.*

We apologize and the statement is now modified.

7. *Use different colors or patterns in graphs in Fig. S1C, E and G. It is hard to differentiate between the two datasets.*

Sorry for the inconvenience and the respective graph patterns are now modified so that it can be easily understood.

Referee # 2:

1- *If I understand correctly the authors' model, it is proposed that upon infection, the miRNA repression will be released to support cytokine expression. In Figure 1, it is clear that the expression level of the IL-6 mRNA is affected upon treatment with LPS. From this set of data, one can argue that the activation of macrophages by LPS does not relieve the miRNA repression but instead increases the miRNA targets transcription leading to an increase of their expression levels. It will be important to convincingly support the authors model to demonstrate that the transcription of endogenous miRNA targets does not solely contribute to the increase of protein levels. From the data presented in Figure S3, it appears that the level of miRNA*

reporters is not affected upon LPS and alpha-amanitin treatment but knowing that these constructs are under control of strong promoters, they are not optimal to rule out any effect on transcription. To measure the effect on transcription, the authors should measure the level of both pre-mRNA and mRNA molecules.

We appreciate the referee's comment. To rule out the possibility that LPS induced transcriptional surge of miRNA target genes is responsible for derepression of miRNA activity in activated macrophage, we transfected *in vitro* transcribed RL miR-122 reporter mRNA into RAW 264.7 cells expressing miR-122 and performed the luciferase assay upon LPS stimulation (data shown in Supplementary Fig. S3B). We observed a similar decrease in miRNA activity and derepression of RL reporter in activated RAW264.7 cells transfected with reporter mRNA that we have observed in reporter plasmid DNA transfected cells. Transcriptional upregulation of cytokine mRNAs may be an additional mechanism by which the activated macrophage ensured optimum expression of cytokines but reversible derepression of miRNA activity certainly augment the proinflammatory response and thus facilitating the cytokine expression in activated macrophages.

- 2. I am not convinced that measuring the slicing activity is a reliable readout of the miRNA binding to Ago2. Upon LPS treatment, post-translational modifications of Ago2 can affect independently its capacity to bind small RNAs and its slicing activity. For this reason, it is suitable to precisely measure instead the amount of miRNAs associated to Argonaute complexes. The authors presented immunoprecipitation data that indicate that the level of miRNA found in the Ago2 complex decrease upon LPS treatment but because others human Agos can also contribute to the miRNA-targeted repression, the interaction of all four human Agos with miRNAs upon LPS treatment should be assessed as well. To circumvent the lack of reliable Argonaute specific antibodies for IP, the authors should perform miRNA pulldown complexes using 2'-O-methylated antisense RNA column with more than one miRNA (at least with one more (miR-122 or miR-16) along with let-*

7a). *In all cases, it will be important to measure the levels of miRNAs in the input fractions to confirm that their levels are not affected upon LPS treatment.*

As mentioned in the next section, Ago2 is the most abundant among all Ago proteins in RAW 264.7 cells where Ago4 expression is very low. So the effect on Ago2 should be the primary contributor in LPS induced derepression of miRNA target genes in RAW 264.7 cells. Additionally, we have performed immunoprecipitation experiment with HA-tagged AGO1 and AGO3 and measured amount of miRNAs associated with immunoprecipitated materials. With LPS stimulation, the miRNA association to HA-AGO3 did not decrease rather increased several fold. Although the Tyr 529 is conserved, Ago3 does not have Ser at 387 position (Rudel S. *et al*, NAR, 2010). Ser 387 is known to be phosphorylated upon p38 MAPK activation. In addition to Tyr 529 phosphorylation, Ser 387 phosphorylation may be required for miRNA loss (Zeng Y, The Biochem J 2008). This may be the reason why miRNA association with Ago3 is not decreased upon LPS stimulation. In case of HA-AGO1, miRNA association also decreased with LPS stimulation (Figure 2F).

3. *As mentioned before, it is known that all four Ago proteins can cause miRNA-specific translational repression. I therefore do not understand why knocking down Ago2 and supplementing cells with AGO2Y529F abrogate the miRNA-targeted repression. It was previously reported that this specific phosphorylation site is also found in other human Argonautes (Rudel et al, NAR, 2011). What are the levels of Ago1, 3 and 4 proteins in macrophages and RAW 264.7 cells? Are they expressed at the same level of Ago2? These important informations should be provided and discussed by the authors.*

Since antibody against Ago1, 3, and 4 proteins are not available, we have performed Real-time PCR to measure the mRNA levels of Ago1, 2, 3 and 4. Ago2 is by far the most abundant and highly expressed in RAW 264.7 cells compared to

other Agos (shown in supplementary Figure S2B). So, Ago2 is most likely plays the key role in miRNA mediated repression in RAW 264.7 cells.

4-Figure 1C and S3C: Because the Northern blots associated to the fractionation experiments show different band intensity, it is difficult to conclude that the association of IL-6 and reporter mRNAs with polysomal fractions is decreased upon cell treatment with LPS. Proper quantification will be required.

In naïve macrophage, IL-6 is expressed in very low level. So the band intensity is different to that of activated macrophage. To circumvent the problem, we have isolated polysomes on a 30% sucrose cushion and performed Real-time quantification of IL-6 mRNA in polysomal and non-polysomal fractions in LPS treated or untreated RAW 264.7 cells. The data is included in Figure 1C. This data shows that there is above 20 fold increase of IL-6 mRNA in polysomal pool of LPS treated RAW 264.7 cells.

5-Figure 4: It is important to show the Ago2 siRNA knockdown does not affect FH-AGO2 and FH-AGO2Y529F expression. Western blots should be provided.

The siRNA was designed against 3'UTR of Ago2. So, siAgo2 should knockdown endogenous Ago2 but not exogenously overexpressed Ago2 without 3'UTR. The western blot of FH-AGO2 and FH-AGO2Y529F in presence of siAgo2 is now shown in Figure 4A which shows that the exogenous Ago2 levels remain unaffected in presence of siAgo2.

6-Figure S4: To support that the interaction of Ago2 with GW182 is also affected upon LPS treatment, quantification should be provided

Quantification data are now added to the western blot in Figure S4.

7-Figure 2B: Is the level of repression obtained with the BoxB system comparable with the repression detected with the reporter carrying the let-7a sites? If is not the case, the way the data are currently presented can be misleading.

The repression of RL-5BoxB reporter is independent of miRNA. In Figure 2B it is shown that unlike RL-3xbulge-let-7a reporter, the repression of RL-5BoxB reporter did not decrease with LPS treatment. Here the comparison is done to demonstrate that miRNA independent repression via AGO2 remains unaffected in activated macrophage cells.

Referee # 3:

While the overall concepts in this paper are interesting, there are some significant issues that need to be addressed with new data to support the claims. First, the proposed mechanism is not overly novel because phosphorylation of Tyrosine 529, as the authors acknowledge, has previously been shown to diminish miRNA binding. While the current study does provide a biological context for this, it would be important to know which kinase, or at least which signaling pathways, are being activated during inflammation and subsequently mediate phosphorylation of Ago2 at Y529. The other issue is the lack of data for one to be able to grasp the breadth of this suppression of the miRNA pathway shortly after macrophage activation. Does it only impact a few miRNAs, or miRNAs in general? This could be addressed using HITS-CLIP before and after LPS treatment to determine which miRNAs and specific mRNA targets are impacted by the proposed mechanism.

LPS stimulation to macrophage cells can activate p38 MAPK, ERK or JNK pathway. In our study, using different inhibitors, we have found that only the p38 MAPK inhibitor can partially block the LPS mediated decrease in miRNA activity. So, it seems that p38 MAPK pathway and yet unknown kinase downstream of p38 MAPK signaling pathway may be involved in phosphorylation of Tyr 529 of Ago2. These data are now included in Figure S4G. We have also shown that, the p38 MAPK inhibitor can partially prevent

the LPS mediated loss of miRNA from Ago2 protein as well as it can prevent the increase in Tyr phosphorylation of Ago2.

We have performed Real-time quantification of some other miRNAs that are reported to be important in immune system, from Ago2 immunoprecipitate materials. All the miRNAs show similar type of decrease in association with Ago2 during LPS stimulation (now shown in Figure 2F and also in supplementary Figure S4C). So, we may say that, LPS stimulation has a general impact on miRNAs binding to Ago2. Therefore we may expect a general loss of miRNA targeted mRNAs association with Ago2 in LPS treated macrophages as evident in results described in Figure 2A.

Other issues

1. The authors include too much data to simply demonstrate the well-known fact that macrophages make inflammatory cytokines when they are activated by PAMPs.

Some data are now removed from Supplementary figures.

2. More work should be done in primary macrophages instead, or in addition to, cell lines to confirm findings in physiologically relevant cell types.

Ago2 immunoprecipitation and associated let-7a quantification was performed with extracts prepared from isolated splenic macrophages from untreated or LPS injected mouse. We observed a similar decrease of let-7a association with Ago2 in splenic macrophage upon LPS treatment (Fig. 2E). We also have performed Ago2 immunoprecipitation and let-7a miRNA quantification from the immunoprecipitates from LPS stimulated THP1 cells, a human monocytic cell line (now shown in Supplementary Figure S4B) and observed similar results.

3. In addition to experiments with loss of function mutations at Y529, the authors should also test the effects of a phosphomimetic at Y529, which should constitutively inhibit miRNA-mediated repression and boost cytokine production.

We have done the suggested experiments and contrary to our expectation, we did not observe a boost in cytokine productions in cells expressing the Ago2Y529E. This may be explained by strong miRNA independent binding of Ago2Y529E to certain cytokine mRNAs like TNF- α that did not get reversed upon LPS treatment (see the Figure after our response to Referee 1 second comment). This may leads to a poor

proinflammatory response in macrophage expressing the FH-AGO2Y529E that prevents stabilization and translatability of other key Ago2 associated cytokine mRNAs in presence of LPS.

4. *During experiments where mutant Ago2 is added to Raw264.7 cells with knocked down endogenous Ago2, how efficient was the transfection of mutant Ago2 constructs? What percentage of the cells are receiving the expression constructs? Raw264.7 cells can be challenging to transfect and the authors should make this determination. Based upon the observed impact on cytokine production, one would have to achieve a fairly high level of transfection.*

Indeed RAW 264.7 is a hard to transfect cell line. We have used Fugene HD (Roche) to transfect RAW 264.7 cells with FH-Ago2 or Y529F mutant. Using a GFP reporter plasmid where is cloned in same vector backbone, the transfection efficiency was fairly good with Fugene HD. It was almost 60% of cells that were green when transfected with the GFP reporter using Fugene HD.

5. *Is it possibly that phosphorylation of Ago2 at Y529 can impact cytokine expression through a mechanism that is independent of the miRNA pathway? For example, there have been some reports of Ago proteins associating with gene promoters in the nucleus. OR, Ago2 might be part of a signaling cascade downstream from TLR signaling.*

To check out such a possibility, we have measured association of Ago2 with cytokine gene promoters following LPS stimulation by Chromatin Immunoprecipitation (ChIP) assay. We found that Ago2 association with TNF- α promoter did not increase rather

decreased with LPS stimulation. By nuclear fractionation study, we have shown that Ago2 level does not increase in nucleus with LPS treatment (Now shown in Supplementary Figure S2G). Moreover, in Fig 2B, we have shown that miRNA independent repression of artificially tethered Ago2 remains unaffected with LPS stimulation. So, we may rule out the possibility that phosphorylation of Ago2 at Y529 can impact cytokine expression through a miRNA independent mechanism.

6. It would also be useful to knockdown Dicer and see if enhanced cytokine production is observed. This would further support a role for miRNAs in suppressing the initial burst of cytokine production by macrophages.

This is an interesting proposition and we have tried the experiment. Upon Dicer1 depletion we have documented an enhanced cytokine production both in naïve and activated macrophage cells (Supplementary Fig. S1G).

7. There are several typos and grammatical mistakes throughout the manuscript that should be corrected.

We have tried to rectify the typos and grammatical mistakes in the revised version as much as possible.

Reference

1. Rudel S, Wang Y, Lenobel R, Korner R, Hsiao HH, Urlaub H, Patel D, Meister G (2011) Phosphorylation of human Argonaute proteins affects small RNA binding. *Nucleic acids research* **39**: 2330-2343
2. McMullen MR, Cocuzzi E, Hatzoglou M, Nagu LE (2003) Chronic ethanol exposure increases the binding of HuR to the TNF alpha 3'-untranslated region in macrophages. *J Biol Chem* **278(40)**: 38333-41
3. Zeng Y, Sankala H, Zhang X, Graves PR (2008) Phosphorylation of Argonaute 2 at serine-387 facilitates its localization to processing bodies. *The Biochemical journal* **413**: 429-436

Thank you for your patience while your study was peer-reviewed once again at EMBO reports. I have taken over the handling of your manuscript because Alejandra Clark, who was my maternity leave cover, is no longer at the journal.

We have now received the enclosed reports from the three original referees. As you will see, although referee 3 is now supportive of publication, referees 1 and 2 have outstanding issues and cannot recommend publication at this stage. Upon further discussion with the referees, it was clear that the evidence for the biological relevance of Ago2 phosphorylation is still not sufficiently conclusive. As mentioned in our previous decision letter, it is EMBO reports policy to allow one round of revision only and, thus, we have no choice but to decline the publication of your manuscript.

Given the potential interest of the findings, we would be open to considering a new manuscript on the same topic if you were to develop the work along the lines indicated by the referees and obtained experimental data to considerably and conclusively strengthen the message of the study. To be completely clear, however, I would like to stress that if you were to send a new manuscript this would be treated as a new submission rather than a revision and would be reviewed afresh at the editorial level, especially with respect to novelty at the time of resubmission and how well the referee concerns were addressed.

I am very sorry that a negative decision is the outcome of such a protracted process, and hope that the referee comments are helpful in your continued work in this area.

REFEREE REPORTS:

Referee #1:

While the revised version of the manuscript is technically improved, some of the conceptual issues remain unanswered.

1. The exact contribution of LPS-induced Ago2 phosphorylation and consequent miRNA target derepression to the inflammatory program is not clear to me. The authors assume that the net effect here will be proinflammatory, but there is very little data to support this conclusion in the manuscript. The data in Fig.4A suggests that AGOY529F mutant has little effect on the induction of IL-10 by LPS and yet upon infection with *Leishmania* macrophages that express AGOY529F mutant output more IL-10 (Fig.4B). Where is the causality here? What happens to proinflammatory cytokine levels in the *Leishmania* infection experiment? What effect AGO2Y529E mutant has on LPS induced cytokine production and *Leishmania* infection? According to Fig.4A, AGO2Y529F mutant has strong effect on the induction of IL1beta by LPS. Which miRNA(s) is behind this effect, since TargetScan predicts no conserved miRNA binding sites for the IL-1beta mRNA?

2. The authors suggest that LPS treatment has no effect on the total miRNA levels (e.g. let-7a levels stay constant in LPS treated cells), yet the levels of all RISC-bound miRNAs decrease by about 50%. So, where do the 50% of mature miRNAs that are expunged from RISC upon Ago2 phosphorylation go? Why are they not degraded? The storage and reloading of single-stranded miRNAs back onto Ago2 as the authors propose is hard to fathom, since loading of miRNAs onto RISC usually happens from a double-stranded miRNA duplex via Dicer/TRBP/PACT complex.

3. LPS treatment leads to release of let-7a from Ago2 as a result of Ago2 phosphorylation, but causes an increase in let-7a associated with Ago3 (Fig. 2F). This result implies that Ago3 is refractory to LPS-induced phosphorylation on Y529, however, since Y529 motif is highly conserved between different Ago proteins, some explanation for this phosphorylation selectivity has to be provided.

4. The significantly increased association of phosphomimetic Ago2 mutant that is capable of little miRNA binding with TNF mRNA (see authors' rebuttal letter) strongly contradicts the proposed model and suggests that Ago2 phosphorylation might confer effects on gene expression that are independent of miRNA. The authors should check the association of a few other cytokine mRNAs (IL-6, IL-1beta, etc) with the AGO2Y529 mutant as well as incorporate this mutant in experiments described in Fig.3E.

5. It is not clear to me why in Fig. 3B analysis of RISC-bound miRNA levels is done by AGO2 IP from 293 cells with the use of exogenous miRNA and RAW264 lysates. I would rather prefer to see this AGO2 IP done in RAW264 complementing the data in Fig.3E and also to extend it to incorporate analysis of endogenous miRNA binding.

Referee #2:

Although the authors made a significant effort to address several concerns raised during the first round of review, there is still some issues that need to be addressed prior to the publication of this manuscript.

One of the main issue is related to the new set of data supporting the contribution of the p38 MAPK pathway in the phosphorylation of Ago2 upon LPS stimulation. I appreciate that the authors now provide experimental evidence for the implication of the p38 MAPK pathway but with this new dataset, it becomes essential (and more logical) to test the contribution of Ago2 phosphorylation of Serine 387 in macrophage activation. It is also relevant to include such analysis because the authors use this observation to justify why the binding of miRNAs to Ago3 is not affected the same way as Ago2 upon LPS stimulation (in contrast to Tyr529, Ser387 is not conserved in Ago3).

I do appreciate that the reviewers addressed experimentally the contribution of miRNA targets transcription upon LPS treatment with a mRNA reporter. The derepression observed upon LPS treatment appears rather modest (at least not at the level observed with reporter plasmids as stated by the authors). A proper statistical analysis should be performed to convince the readers that the miRNA-mediated repression of the mRNA reporter is also relief upon LPS treatment.

Referee #3:

All of my concerns have been adequately addressed.

Resubmission - authors' response

15 July 2013

Ref: Resubmission of the **New Version of Manuscript 36840** by Mazumder et al. to EMBO reports.

Thank you very much for handling our manuscript and we appreciate your decision to reconsider the Manuscript 36840 by Mazumder et al. for EMBO reports. We are submitting the new revised version of the manuscript titled “**A transient reversal of miRNA-mediated repression controls macrophage activation**” and we have done several additional experiments to address all specific concerns that the reviewers 1 and 2 had with the previous versions of this manuscript.

After the 2nd round of review, we are delighted to find that reviewer 3 is satisfied with the changes in the revised version. While the Reviewer 2, quite logically, has raised couple of issues, Referee 1 has pointed out several new concerns. This is surprising for us as few of the raised issues are against findings that were already there in the initially submitted version of this manuscript and are not raised after the 1st round of review! This caused us to reschedule the work and to do repetitive

experiments what could be easily completed and included in the previous version of this manuscript. Nonetheless we appreciate the concerns both the reviewers have.

With this revised version we have several new panels in Fig. 3 and 4 and also in Supplementary Fig S2, 3 and 4. We additionally have one additional supplementary Fig S5 and two supplementary movies with this version. Having additional experiments incorporated, I hope this manuscript is now sufficiently convincing to show that the derepression of miRNA-mediated gene repression is the key feature of macrophage activation and is essential for proinflammatory response in mammalian macrophage upon LPS treatment. Please find the file attached containing a detailed point-by-point response to the Referees' comments.

At last but not least, we would like to thank Reviewers for their constructive criticism that prompted us to explore some new aspects of this phenomenon otherwise we could have missed. Thank you also for your kind consideration and for your interest to reconsider the revised version. We hope you will find it suitable now for publication in EMBO Reports.

Answers to Referee's Comments

Referee 1

While the revised version of the manuscript is technically improved, some of the conceptual issues remain unanswered.

1. The exact contribution of LPS-induced Ago2 phosphorylation and consequent miRNA target derepression to the inflammatory program is not clear to me. The authors assume that the net effect here will be proinflammatory, but there is very little data to support this conclusion in the manuscript. The data in Fig.4A suggests that AGOY529F mutant has little effect on the induction of IL-10 by LPS and yet upon infection with Leishmania macrophages that express AGOY529F mutant output more IL-10 (Fig.4B). Where is the causality here? What happens to proinflammatory cytokine levels in the Leishmania infection experiment? What effect AGO2Y529E mutant has on LPS induced cytokine production and Leishmania infection? According to Fig.4A, AGO2Y529F mutant has strong effect on the induction of IL1beta by LPS. Which miRNA(s) is behind this effect, since TargetScan predicts no conserved miRNA binding sites for the IL-1beta mRNA?

LPS induced proinflammatory response in mammalian macrophages is characterized by increased expression of proinflammatory cytokines like IL-1beta, IL-6 and TNF- α and is associated with an opposite or no effect on IL-10 level, a key anti-inflammatory cytokine in mammalian macrophage. We have found this response is opposed in cells expressing the AGO2 mutant that is defective to loose miRNA in response to LPS stimulation (Fig 4A).

Interestingly, in response to an insult with a high concentration of a pathogen-derived factor, macrophage induces an elevated proinflammatory response by expressing excessive TNF- α , ROS and NO. This sudden surge of proinflammatory cytokines and ROS production also induces apoptotic death of macrophage (Xaus J, *et al*, 2000). In response to a high concentration of LPS, RAW 264.7 cells expressing FH-AGO2Y529F showed reduced apoptotic death evident by TUNEL assay and Caspase-3 and-9 cleavage (See below). This was consistent with a reduced amount of proinflammatory cytokines produced in AGO2 mutant expressing RAW264.7 cells. We do not incorporate this data in the main manuscript as we consider this would make the manuscript partly defocused.

Fig. 1 Prolonged LPS exposure failed to induce apoptosis in RAW 264.7 cells expressing FH-AGO2Y529F. TUNEL assays were done to detect apoptotic cells in FH-AGO2 or FH-AGO2Y529F expressing RAW 264.7 cells during LPS treatment (100ng/ml) (A). Apoptosis level was also monitored by checking the level of cleaved Caspase-3 and -9 detected by western blot using cleaved caspase-3 and -9 specific antibodies. beta-Actin was used as loading control (B).

On the contrary to LPS that induces proinflammatory response, Leishmania parasites prevent the production of proinflammatory cytokines and induce anti-inflammatory immune response for its survival inside the macrophage cells. As reported earlier, induction of IL-10 is a hallmark of *L.donovani* infection of macrophage. In the experiment described in Fig 4B, the FH-AGO2 or FH-AGO2Y529F or FH-AGO2Y529E expressing RAW 264.7 cells were treated with LPS for 4h, followed by infection with *L.donovani* for 24h. This was done to induce a proinflammatory response before the cells interaction with the parasite. If the preincubation can induce an increase of proinflammatory cytokine production, this should prevent infection and make the pre-treated macrophage immune to Leishmania infection. Upon injection *L. donovani* switches the proinflammatory response to an anti-inflammatory one and increases IL-10 production in macrophage. As the Y529F mutant expressing macrophages produced lesser pro-inflammatory cytokines on exposure to LPS, they were more prone to infection. Higher Leishmania infection induced higher IL-10 cytokines as shown in Fig 4B. But the FH-AGO2 Y529E over-expressing RAW 264.7 cells showed no significant difference in IL-10 production compared to FH-AGO2 (Fig 4B).

Similarly, The pro-inflammatory cytokines TNF- α and IL-6 decreased more in LPS pre-treated, Leishmania infected RAW 264.7 cells expressing Y529F mutant compared to FH-AGO2 expressing cells. The phosphomimetic mutant FH-AGO2Y529E mutant did not show any significant difference in TNF- α and IL-6 production in comparison to FH-AGO2 under those conditions.

We took 3'UTR of mouse IL-1beta mRNA and search for miRNA target sites in miRBase, it showed IL-1beta3'UTR does have several miRNA binding sites e.g. let-7f, miR-361, miR-4661 etc. That's could explain why IL-1beta was also showing similar trend and is consistent with observed reduced binding of IL-1beta mRNA with FH-AGO2 in LPS treated cells (see below)

Fig. 2 Relative binding of IL-1b mRNA with AGO2 and its mutant in LPS-treated RAW264.7 cells.

2. The authors suggest that LPS treatment has no effect on the total miRNA levels (e.g. *let-7a* levels stay constant in LPS treated cells), yet the levels of all RISC-bound miRNAs decrease by about 50%. So, where do the 50% of mature miRNAs that are expunged from RISC upon Ago2 phosphorylation go? Why are they not degraded? The storage and reloading of single-stranded miRNAs back onto Ago2 as the authors propose is hard to fathom, since loading of miRNAs onto RISC usually happens from a double-stranded miRNA duplex via Dicer/TRBP/PACT complex.

'The storage and reloading of single-stranded miRNA back onto dephosphorylated Ago2' is a possible hypothesis to explain reversal of miRNA activity in LPS treated cells. Although reloading of pre-existing miRNAs back to dephosphorylated Ago2 can explain the results, till date there is no such evidence of single stranded miRNA loading to AGO2 happens *in vivo*. For certain miRNAs, transcriptional surge in LPS treated cells can also compensate the loss of Ago2-uncoupled miRNAs in late phase of activation where *de novo* double stranded miRNAs may be processed and reloaded to dephosphorylated Ago2 when the unloaded single stranded miRNAs are lost. But with siRNAs or miRNA-122 mimic transfected cells where the input miRNA and siRNA levels could not be affected by *de novo* transcription of its precursors, reloading of uncoupled miRNAs is certainly the only probable option to explain the regained the repressive activity upon prolonged LPS-treatment of macrophage cells. To investigate the exact sub-cellular localization of unloaded miRNAs, we did an OptiprepTM density gradient analysis of naive and LPS stimulated macrophage and checked the distribution of miR-122 in RAW264.7 cells with different cellular organelles. From the present data it seems that a large fraction of miRNAs were located in Lysozomes but remained protected from degradation. The exact mechanism of this protection is not clear but we have also found, with LPS treatment, the miR-122 mimic were expunged from multivesicular body (MVB)/endosome enriched fractions and accumulated to ribosome enriched fraction possibly for reloading to form new miRNPs (Fig S3E). This is certainly a very interesting observation for our future exploration of the mechanism of miRNA reloading prolonged LPS treated macrophage and we would like to thank the Referee for that as his comments prompted us to do this experiment.

3. LPS treatment leads to release of *let-7a* from Ago2 as a result of Ago2 phosphorylation, but causes an increase in *let-7a* associated with Ago3 (Fig. 2F). This result implies that Ago3 is refractory to LPS-induced phosphorylation on Y529, however, since Y529 motif is highly conserved between different Ago proteins, some explanation for this phosphorylation selectivity has to be provided.

Although, we don't have any concrete evidence why Ago3 is refractory to LPS-induced

phosphorylation on Y529, but from an Optiprep gradient analysis we have seen that the suborganelle localization of HA-AGO3 is different from that of HA-AGO2 in LPS treated RAW264.7 cells. Unlike FH-AGO2, FH-AGO3 is predominantly located with late endosome fraction and shows very little and no association with fractions enriched for ribosomes (Fig S5H; Fig. S3E). Moreover, from the homology modelling data, we have found that the cavity size of the ATP binding pocket is different in AGO3 than AGO2. These may have a role in their interaction with the putative kinase(s), and observed differences in Y529 phosphorylation and miRNA association of Ago3 during LPS stimulation (Fig S5B, Supplemental Movie 1-2). Lastly Ago3 has very low expression in RAW264.7 cells and therefore could not function as a key player in miRNA mediated repression.

Please also see our reply to comments of Referee 2.

4. *The significantly increased association of phosphomimetic Ago2 mutant that is capable of little miRNA binding with TNF mRNA (see authors' rebuttal letter) strongly contradicts the proposed model and suggests that Ago2 phosphorylation might confer effects on gene expression that are independent of miRNA. The authors should check the association of a few other cytokine mRNAs (IL-6, IL-1beta, etc) with the AGO2Y529 mutant as well as incorporate this mutant in experiments described in Fig.3E.*

In our previous response to Referees' comments we have tried to explain why TNF- α behaves differently with AGO2Y529E mutant. Here, the Refree has tried to suggest that the phosphorylation of Ago2 affects gene expression in a miRNA independent manner. If this argument is true we should expect to have a strong surge in expression of proinflammatory cytokines in RAW264.7 cells expressing this phosphormimetic mutant of AGO2 and upon LPS priming it should show a strong proinflammatory response and prevent *L.donovani* infection and subsequent anti-inflammatory response in macrophages encountering the parasites (please also see our reply against point 1 and Fig. 4B-C). This indeed was not the case. Neither any of the proinflammatory cytokines showed increased expression in FH-AGO2Y529E mutant expression cells, neither cells expressing this mutant showed any difference in LPS induced derepression of miRNA target messages upon LPS exposure compared to FH-AGO2 expressing cells. We have checked the IL-6 mRNA association with FH-AGO2, FH-AGO2Y529F and FH-AGO2Y529E in LPS stimulated macrophages and found higher level of IL-6 mRNA remained bound to Y529F mutant even after LPS treatment. Whereas, the phosphomimetic Y529E mutant showed lesser association with IL-6 (Figure shown below). We have also done the experiment described in Fig 3E with Y529E mutant and incorporated the data in Fig 3E. RAW 264.7 cells expressing FHA-AGO2Y529E did not show any significant difference in miRNA activity compared to wild type AGO2 upon LPS stimulation. These results strongly suggest that the LPS induced phosphorylation of Ago2 and associated loss of miRNA are both necessary for observed proinflammatory response in LPS-treated mammalian macrophage and the phosphorylated AGO2 can't have the effect in a miRNA independent manner.

Fig. 3 Amount of IL-6 mRNA bound to AGO2 and its mutant forms before and after LPS stimulation.

5. It is not clear to me why in Fig. 3B analysis of RISC-bound miRNA levels is done by AGO2 IP from 293 cells with the use of exogenous miRNA and RAW264 lysates. I would rather prefer to see this AGO2 IP done in RAW264 complementing the data in Fig.3E and also to extend it to incorporate analysis of endogenous miRNA binding.

In Fig 3B, analysis of RISC-bound miRNA levels was done by HA-AGO2 IP from LPS stimulated RAW 264.7 cells expressing FH-AGO2 or Y529F mutant. In Fig 3E, luciferase assays were done using RL reporter having binding sites for miR-122. So, we measured exogenously expressed miR-122 level in Fig 3B.

We have also measured different endogenous miRNA levels associated with either FH-AGO2 or the Y529F mutant in naïve and LPS activated macrophage and incorporated the data in Fig 3B. All the miRNA association to Y529F mutant were insensitive to LPS stimulation.

Referee #2:

Although the authors made a significant effort to address several concerns raised during the first round of review, there is still some issues that need to be addressed prior to the publication of this manuscript.

One of the main issue is related to the new set of data supporting the contribution of the p38 MAPK pathway in the phosphorylation of Ago2 upon LPS stimulation. I appreciate that the authors now provide experimental evidence for the implication of the p38 MAPK pathway but with this new dataset, it becomes essential (and more logical) to test the contribution of Ago2 phosphorylation of Serine 387 in macrophage activation. It is also relevant to include such analysis because the authors use this observation to justify why the binding of miRNAs to Ago3 is not affected the same way as Ago2 upon LPS stimulation (in contrast to Tyr529, Ser387 is not conserved in Ago3).

We have measured the Ser phosphorylation level of Ago2 by western blotting immunoprecipitated Ago2 using anti-phosphoserine antibody. Phosphorylation level did not found to be increased rather decreased with LPS stimulation (Fig S4G). From our homology modelling it becomes apparent that S387 is located far away from Y529 to have a direct effect of S387 phosphorylation on Y529 phosphorylation. Additionally, recent evidence suggests phosphorylation of S387 has been implicated in increased miRISC activity (Horman et al, 2013) which is opposite to what we saw in LPS treated RAW264.7 cells. S387 phosphorylation increases P-body localization of AGO2 what is also opposite to our observation and we saw a decreased P-body localization of Ago2 in LPS treated macrophages was noted. Considering these facts it is unlikely that Ser 387 should have an effect on Y529 phosphorylation and subsequent loss of miRNA from phosphorylated Ago2. So what we hypothesize previously that phosphorylation at S387 might have any role in phosphorylation in Y529 may not be correct and we have rectified the text accordingly. Regarding effect of LPS on Ago3 we have already answered the question in our response to Referee 1 comment 3.

I do appreciate that the reviewers addressed experimentally the contribution of miRNA targets transcription upon LPS treatment with a mRNA reporter. The derepression observed upon LPS treatment appears rather modest (at least not at the level observed with reporter plasmids as stated by the authors). A proper statistical analysis should be performed to convince the readers that the miRNA-mediated repression of the mRNA reporter is also relief upon LPS treatment.

We appreciate the referee's comment. We have measured the reporter mRNA levels used in the luciferase assay and normalized the luciferase values against the respective mRNA level. It shows 40% loss of miRNA activity with LPS treatment. We have also quantified the amount of target mRNA bound by Ago2 in RAW264.7 cells transfected with reporter mRNAs and treated with LPS. There is a similar decrease in mRNA association with Ago2 in LPS treated cells (Fig S3C).

Referee #3:

All of my concerns have been adequately addressed.

We are very happy to know that.

References

Horman SR, Janas MM, Litterst C, Wang B, MacRae IJ, Sever MJ, Morrissey DV, Graves P, Luo B, Umesalma S, Qi HH, Miraglia LJ, Novina CD, Orth AP (2013) Akt-mediated phosphorylation of argonaute 2 downregulates cleavage and upregulates translational repression of MicroRNA targets. *Molecular cell* **50**: 356-367

Xaus J, *et al.* (2000) LPS induces apoptosis in macrophages mostly through the autocrine production of TNF-alpha. *Blood* 95(12):3823-3831.

3rd Editorial Decision

06 August 2013

Thank you for your patience during the peer-review of your resubmission to EMBO reports. It has now been seen by referees 1 and 2 of the previous version, whose comments are pasted below. As you will see, referee 2 is now essentially supportive of publication although recommends careful editing of the text, whereas referee 1 is not satisfied with the proof of physiological relevance. Upon further discussion within the editorial team and with referee 2, we feel that at this point it would be outside of the scope of this study to ask you to change the infection model. In addition, your current study focuses on the initial stages of Leishmania infection, whereas your recent publication mentioned by referee 1 dealt with later stages. Nevertheless, additional discussion is necessary to clarify this point and contextualize the results.

On the other hand, another important aspect to address during revision is that of length of the main text, which is over 6,000 characters above our 30,000 upper limit. Shortening will be made easier by reformatting the reference section to EMBO reports style, which uses numbers as in-text citations. Nevertheless, you will also have to go through the text and more succinctly summarize your findings were appropriate without affecting overall readability and comprehension (and adding the necessary discussion mentioned above).

In addition, please note that basic Materials and Methods required for understanding all the experiments performed must remain in the main text, although additional detailed information may be included as Supplementary Material. In the present version it seems as though two full subsections have been chosen to remain in the main text whereas all other relevant information is supplementary. Instead, an abbreviated version of the most relevant subsections has to be in the main text, and extended versions can be supplementary. Importantly, the subsection on statistical analyses of the data has to be included in the main text. In this regard, please ensure that the number of independent experiments used to calculate errors is indicated in all figure legends, as now the legends to most panels are missing this information. The error bars seem unusually small; were they calculated from replicates within an experiment or from independent experiments? Please note that it is not appropriate to present error bars of replicates within a single experiment, and that at least

three independent experiments should be used for their calculation. I would refer you to the following articles for guidance: Vaux et al., EMBO rep 2012; Cumming et al. JCB 2007. Please also indicate in the figure legend whether calculations were made from independent experiments.

I look forward to seeing a final version of your manuscript when it is ready.

REFEREE REPORTS:

Referee #1:

The revised version of the manuscript has reasonably addressed some of my previous reservations. However, I am still concerned that Leishmania infection model that the authors use to demonstrate the physiological relevance of the inflammation-induced Ago2 phosphorylation (Fig. 4B-E) is too convoluted and its results are a bit too difficult to interpret. Ago2 mutants show very modest if at all effects on the proinflammatory cytokine induction by LPS (Fig. 4A,C) and it is only after Leishmania addition that some moderate effects on TNF and IL-6 production can be observed in this model (Fig. 4C). Besides, as was recently published by the same group, Leishmania donovani can affect global miRNA expression by gp63-mediated proteolytical cleavage of Dicer in mouse hepatocytes and macrophages (Ghosh, J et al. Cell Host Microbe 2013 13:277-88). This ability of Leishmania donovani to interfere with miRNA biogenesis makes the interpretation of the experiment difficult and I suggest that the authors try to prove their point here by using a different biological model.

Referee #2:

The authors appropriately addressed all my comments with this new revised version. Before publication, it will be important to carefully proof read the text and figures. Some errors/discrepancies particularly in the nomenclature are still found (AGO2 vs Ago2, let-7a vs let7a...).

3rd Revision - authors' response

19 August 2013

Thank you very much for your previous email to let us know the decision of considering the revised version of the manuscript by Mazumder et al for publication in *EMBO reports*. We are very happy to know this decision. We have now prepared the final version of this manuscript as per your recommendation.

We have now added a section in appropriate place in the Results and Discussion part to address the importance of regulation of inflammatory response by miRNAs in the context of *L. donovani* infection of mammalian macrophage.

We have introduced error bars only for experiments where results from minimum three experimental replicates are available and have mentioned the “n” numbers in the figure legends. We have also uploaded raw data files for most of the Western and Northern blots data with sufficient annotation as per the guidelines. Additionally we preferred to include a couple of MS Excel data set for two most important experiments along with enlarged version of few microscopic images for readers benefit.

We also have reduced the text to 31,000 characters (approx) and that without compromising the readability and clarity of the manuscript. We have reshuffled the Methods section

also to accommodate concise versions of most important experimental procedures in the main text while keeping the detail procedure available online as Supplementary Information.

We also have tried our best to rectify all grammatical and typological and nomenclature related discrepancies while finalizing the text during editing. We hope now the Final version of this manuscript with these revisions is acceptable for publication in *EMBO reports*.

4th Editorial Decision

22 August 2013

I have taken over the handling of your manuscript in order to speed up the process, as my colleague Nonia is currently not in the office. I am very pleased to accept your manuscript for publication in the next available issue of EMBO reports. Thank you for your contribution to our journal.

Nonia suggested some changes to the abstract, as follows. Can you please let us know whether you agree with these changes? Thank you very much.

Abstract:

In mammalian macrophages, the expression of a number of cytokines is regulated by miRNAs. Upon macrophage activation, proinflammatory cytokine mRNAs are translated, although the expression of miRNAs targeting these mRNAs remains largely unaltered. We show that there is a transient reversal of miRNA-mediated repression during the early phase of the inflammatory response in macrophages, which leads to protection of cytokine mRNAs from miRNA-mediated repression. This de-repression occurs through Ago2 phosphorylation, which results in its impaired binding to miRNAs and to the corresponding target mRNAs. Macrophages expressing a mutant, non-phosphorylatable Ago2-which remains bound to miRNAs during macrophage activation-have a weakened inflammatory response and fail to prevent parasite invasion. These findings highlight the relevance of the transient relief of miRNA repression for macrophage function.

As part of the EMBO publication's Transparent Editorial Process, EMBO reports publishes online a Review Process File to accompany accepted manuscripts. As you are aware, this File will be published in conjunction with your paper and will include the referee reports, your point-by-point response and all pertinent correspondence relating to the manuscript.

If you do NOT want this File to be published, please inform the editorial office within 2 days, if you have not done so already, otherwise the File will be published by default [contact: emboreports@embo.org]. If you do opt out, the Review Process File link will point to the following statement: "No Review Process File is available with this article, as the authors have chosen not to make the review process public in this case."

Thank you again for your contribution to EMBO reports and congratulations on a successful publication. Please consider us again in the future for your most exciting work.